# Backward Walking Induces Significantly Larger Upper-Mu-Rhythm Suppression Effects Than Forward Walking Does

**DOI:** 10.3390/s20247250

**Published:** 2020-12-17

**Authors:** Nan-Hung Lin, Chin-Hsuan Liu, Posen Lee, Lan-Yuen Guo, Jia-Li Sung, Chen-Wen Yen, Lih-Jiun Liaw

**Affiliations:** 1Department of Mechanical and Electro-Mechanical Engineering, National Sun Yat-Sen University, Kaohsiung 80424, Taiwan; d983020009@student.nsysu.edu.tw (N.-H.L.); d033020003@student.nsysu.edu.tw (J.-L.S.); 2Department of Occupational Therapy, Kaohsiung Municipal Kai-Syuan Psychiatric Hospital, Kaohsiung 82445, Taiwan; isu6a394@cloud.isu.edu.tw; 3Department of Occupational Therapy, I-Shou University, Kaohsiung 82445, Taiwan; posenlee@isu.edu.tw; 4Department of Sports Medicine, Kaohsiung Medical University, Kaohsiung 80708, Taiwan; yuen@kmu.edu.tw; 5Department of Physical Therapy, College of Health Science, Kaohsiung Medical University, Kaohsiung 80708, Taiwan; 6Neuroscience Research Center, Kaohsiung Medical University, Kaohsiung 80708, Taiwan; 7Department of Physical Medicine and Rehabilitation, Kaohsiung Medical University Hospital, Kaohsiung 80708, Taiwan; 8Department of Medical Research, Kaohsiung Medical University Hospital, Kaohsiung 80708, Taiwan

**Keywords:** movement-related cortical potentials, treadmill, backward walking, electroencephalography rhythms, gait rehabilitation

## Abstract

Studies have compared the differences and similarities between backward walking and forward walking, and demonstrated the potential of backward walking for gait rehabilitation. However, current evidence supporting the benefits of backward walking over forward walking remains inconclusive. Considering the proven association between gait and the cerebral cortex, we used electroencephalograms (EEG) to differentiate the effects of backward walking and forward walking on cortical activities, by comparing the sensorimotor rhythm (8–12 Hz, also called mu rhythm) of EEG signals. A systematic signal procedure was used to eliminate the motion artifacts induced by walking to safeguard EEG signal fidelity. Statistical test results of our experimental data demonstrated that walking motions significantly suppressed mu rhythm. Moreover, backward walking exhibited significantly larger upper mu rhythm (10–12 Hz) suppression effects than forward walking did. This finding implies that backward walking induces more sensorimotor cortex activity than forward walking does, and provides a basis to support the potential benefits of backward walking over forward walking. By monitoring the upper mu rhythm throughout the rehabilitation process, medical experts can adaptively adjust the intensity and duration of each walking training session to improve the efficacy of a walking ability recovery program.

## 1. Background

This section is organized as follows: Section 1.1 reviews the similarities and differences between forward walking (FW) and backward walking (BW), Section 1.2 discusses brain response measurement methods, and Section 1.3 introduces the brain response measures employed in this study.

### 1.1. Similarities and Differences between FW and BW

As an option for physical exercise and clinical rehabilitation, preliminary results have suggested that backward walking (BW) can offer additional advantages over forward walking (FW). However, as indicated by several review papers, current evidence regarding the benefits of BW over FW is inconclusive [1,2,3,4].

Fundamentally, BW is merely a temporal reversal of FW [5,6], kinematically, the joint angle waveforms in BW gait are essentially time-reversed, relative to the corresponding waveforms in FW gait [7]. In addition, kinetically, BW has similar muscle activation patterns as FW but with the temporal cycling of muscle contraction reversed [8].

BW and FW also have many physiological and biomechanical differences. Specifically, compared to FW, BW elicits greater cardiorespiratory, metabolic, and perceptual responses [9,10], has higher mean electromyogram activity [7], reduces the angular dispersion of the spine segments around the roll axis and the lateral oscillations [11], has stronger cortical response [12], elicits greater activation within medial sensorimotor cortices [13], demands additional attentional resources [14], and requires significantly lower ankle power [15].

Previous studies have also demonstrated the potential of BW for rehabilitation. For example, a comparison of FW and BW training results suggested that static balance and muscle strength, particularly in the quadriceps and ankle plantar flexor, improved with BW. BW was also considered a safe training method for hamstring strain rehabilitation because it involves relatively low eccentric loading on the hamstring muscle group [16]. Treadmill training results revealed that BW was more effective than FW for improving anaerobic and balance performances [17,18].

In summary, compared with FW, studies have demonstrated the potential of BW for physical training and rehabilitation. However, to the best of our knowledge, no previous study has attempted to use EEG to compare the effects of FW and BW. Although we still don’t know which is the cause and which is the effect, a clear association between walking and the *μ*-rhythm suppression effects has been reported in the literature. By comparing the *μ*-rhythm suppression effects of FW and BW, and by validating these comparative results with previous *μ*-rhythm suppression effect studies, it is hoped that this work can shed some light on the interactions between the *μ*-rhythm suppression effects, walking directions, and rehabilitation outcome.

### 1.2. Assessing Cortical Activation by Using EEG Measurements

The planning and execution of walking involves several cortical and subcortical areas [19,20]. The activation of the brain areas, due to motor planning and execution, can be measured using several image and signal acquisition methods, including magnetoencephalography [21,22], functional magnetic resonance imaging [23,24], functional near-infrared spectroscopy [25,26], and EEG [27,28]. Among these modalities, EEG has several distinct advantages in assessing brain responses to locomotion. Firstly, EEG sensors are sufficiently lightweight to wear during locomotion. Secondly, considering the time scale of natural gait, EEG has a sufficient time resolution for measuring gait-related responses. However, measuring cognitive brain dynamics with EEG during real-world behaviors has been historically challenging because EEG measurements are prone to artifacts arising from confounding electrophysiological signals (e.g., signals from the eyes, muscles, and heart), electrode and cable motions, and environmental electrical noise [29,30]. Nevertheless, when artifacts can be effectively suppressed, the genesis of motor planning and execution can be assessed by EEG, and provide insights into brain dynamics [31,32].

A straightforward solution for avoiding these motion artifacts is to only compare the EEG recordings before and after gait experiments [33]. However, this strategy eliminates the possibility of studying brain responses during walking and running periods. Blind source separation techniques, such as independent component analysis (ICA), have also been used to eliminate eye, muscle, and motion artifacts from EEG data [34,35,36,37,38]. However, blind-source-separation-based artifact removal was reported to be inadequate because gait-related movement artifacts remained in many, if not most, of the independent components [39]. Consequently, a stride-order-based moving average artifact template was used to eliminate gate-locked artifacts in a previous study [39]. However, this moving average technique is unsuitable for this study because it also eliminates gait-related brain activities.

Dual-layer EEG hardware, that simultaneously records electrocortical signals and isolated noise from secondary sensors, represents a novel artifact removal approach for EEG recording during locomotion [40,41,42]. A dual-EEG array enables the application of ICA to EEG data obtained from primary scalp electrodes, and noise recordings obtained from noise electrodes, which then enables the recovery of artifact-free ground truth EEG signals. When implementing the aforementioned method, the number of noise electrodes should be ideally equal to the number of primary electrodes. The addition of the noise electrodes inevitably increases the complexity and cost of the measurement system. This problem can be partially mitigated using relatively few noise electrodes. However, such a simplification compromises the efficacy of the approach because movement artifacts vary considerably with electrode location [43].

Instead of using one noise electrode for each EEG channel, this study developed an adaptive filter for each EEG channel. By using a force-platform-instrumented treadmill to measure the ground reaction force (GRF) during gait and by using the GRF as the input, the adaptive filters can be trained to eliminate artifacts from the EEG data. Details of the force-treadmill-based adaptive filtering approach are provided in Section 2.

### 1.3. Mu Rhythm

To differentiate the oscillatory activities of the brain’s neural networks, EEG signal components have been divided into different frequency bands corresponding to distinct oscillatory coupling patterns of neuronal assemblies [44]. In the case of EEG signal components with a frequency of 8–12 Hz, alpha rhythm is considered the fundamental functional operator of the brain for signal processing and communication in sensory or cognitive processes [45,46].

Notably, several variants of alpha rhythm exist. One such variant is the sensorimotor alpha rhythm, which is most apparent in the sensorimotor cortex area and includes the primary motor and primary somatosensory cortices [45,47,48,49,50]. The sensorimotor alpha rhythm has also been referred to as mu rhythm because of its arch-shaped wave morphology that resembles the Greek letter *μ*. The sensorimotor cortex, which includes the primary motor cortex (Brodmann area 4) and the primary somatic sensory cortex (Brodmann area 3). These areas, whose activities can be assessed by C3 and C4 electrodes of the international 10–20 system for EEG electrode placement, are very important for rehabilitation. it was shown that the motor learning process and associated variability promote plasticity in the sensorimotor networks and adjust both motor and perceptual skills [51,52,53] and walking can activate the sensorimotor area [36,42,54,55,56,57].

Although many questions regarding the functionalities and influences of mu rhythm remain unanswered, three mu-rhythm properties are widely supported by the literature and thus deserve special attention. Firstly, mu rhythm has been reported to decrease in amplitude (desynchronize) during cognition and motor tasks. In particular, consistent with its characterization as an idling rhythm, mu rhythm is typically maximal over the sensorimotor cortex when the individual is at rest and is suppressed (desynchronized) during motor-related tasks, including actual and imagery movements [48,58].

Secondly, mu rhythm’s lower (8–10 Hz) and upper (10–12 Hz) frequency components are functionally different from each other with the lower and upper components being non-task-specific and task-specific, respectively [47,59,60,61]. In specific, the lower frequency component reflects a widespread, movement-type, non-specific suppression pattern, whereas the upper frequency component shows a more focused and movement-type specific pattern [48]. These results suggest that *μ*_1_ rhythm and *μ*_2_ rhythm should be independently studied when needed. To independently study these frequency components, the mu-rhythm components in the 8–12-, 8–10-, and 10–12-Hz frequency sub-bands are referred to as *μ*_0_ rhythm, *μ*_1_ rhythm, and *μ*_2_ rhythm, respectively [62,63]. In other words, *μ*_0_, *μ*_1_, and *μ*_2_ rhythms represent the sensorimotor alpha, lower alpha, and upper alpha rhythms, respectively. Moreover, the term *μ*-rhythm represents the union of *μ*_0_ rhythm, *μ*_1_ rhythm, and *μ*_2_ rhythm.

During the execution of different movement types, *μ*_1_ rhythm exhibits a widespread movement-type, nonspecific activity pattern, and is thus relatively insensitive to the movement pattern. As a result, μ1 rhythm is suggested to be related to general attentional processes [64]. By contrast, *μ*_2_ rhythm demonstrates a movement-type specific pattern, which suggests its responses are movement-type dependent [59,64]. Based on these observations, it is believed that *μ*_1_ rhythm probably reflects general task demands and attentional processes and is non-task-specific whereas *μ*_2_ rhythm is task-specific [50].

Thirdly, the degree of *μ*-rhythm suppression is task- and complexity-dependent. For example, *μ*-rhythm suppression effects were found to differ between finger and foot movements [47]. Similarly, the *μ*-rhythm desynchronization effects of an active motor task are larger than those of a passive motor task [65]. Moreover, studies have revealed that increasing task complexity results in a high level of suppression [45,48].

To investigate the potential of BW for gait rehabilitation, the electroencephalogram (EEG) signal responses, induced by FW and BW, were compared by performing treadmill walking experiments. Considering the association between sensorimotor area activation and *μ*-rhythm desynchronization, we evaluated and compared the effects of FW and BW on *μ*-rhythm suppression. The hypotheses of this study are that the *μ*-rhythm (sensorimotor alpha rhythm) suppression effects associated with BW are different from those associated with FW. Since walking can activate the sensorimotor cortex which is an important area for rehabilitation, understanding the differences between the *μ*-rhythm suppression effects of FW and BW may enable the effective planning and implementation of BW training programs for better rehabilitation outcomes. To achieve this goal, we verified the validity of our FW and BW experimental results by considering whether our EEG measurement data agreed with the three aforementioned major *μ*-rhythm properties. Next, we quantitatively and qualitatively compared the effects of FW and BW on *μ*-rhythm suppression.

## 2. Methods

### 2.1. Participants

Nineteen healthy participants (18 men and 1 woman) aged between 22 and 31 years were recruited. None of them had pathological conditions that would compromise their postural or walking ability. All experimental procedures were approved by the Institutional Review Board of the Kaohsiung Medical University Chung-Ho Memorial Hospital, Kaohsiung, Taiwan. Approved number KMUHIRB-E(I)-20170033

### 2.2. Experimental Setup and Protocol

This study employed a force-platform-instrumented treadmill (commonly called a force treadmill) for experimental data collection (Figure 1). Constructed by installing force transducers underneath a standard exercise treadmill, this force treadmill can measure the GRF generated during walking and standing. To measure the GRF, analog voltage signals obtained by the four load cells were first amplified and converted to digital signals by using a 24-bit data acquisition card (NI9234, National Instrument, Austin, TX, USA). With a sampling frequency of 1024 Hz, the digitized force signals were transmitted to a computer by using a universal serial bus chassis (NI cDAQ-9174, National Instrument, Austin, TX, USA) and then low-pass filtered by a 20th-order phase distortionless Butterworth filter with a cutoff frequency of 150 Hz. The accuracy and repeatability of the setup employed in this force treadmill were rigorously tested [66].

During the experiment, in addition to the GRF signal, we also simultaneously recorded the EEG and electrocardiogram (ECG) signals. We used the Texas Instruments ADS1299 EEG Front-End Performance Demonstration Kit (TI-ADS1299EEG-FE) to measure EEG signals. With low power consumption and low input referred noise, ADS1299 is a low-cost EEG measurement device. Its measurement accuracy has been shown to be comparable to that of a laboratory-based EEG system [67]. Eight Ag/AgCl wet electrodes were attached at F3, F4, C3, C4, P3, P4, O1, and O2, as specified by the International 10–20 system [68]. Cz was selected as the reference electrode. The ECG measurement device was an in-house-developed system. The sampling rate was defined as 250 Hz for both EEG and ECG signals. To limit motion artifacts, the EEG wires were enclosed in a flexible protection tube and fastened to the participant’s waist. A custom program, which was written using LabVIEW (National Instruments, Texas, USA), was used for data acquisition. In preparing this program, special attention was given to synchronize the clocks for EEG, ECG, and GRF signals.

At the front and rear ends of the treadmill, two tennis balls were suspended from the ceiling. The height of these balls was adjusted to the height of the participant’s eyes. Participants were instructed to adhere to the following requirements during the experiment: minimize chewing and blinking movements, look straight ahead at the tennis ball, and maintain a constant distance between the body and the tennis ball.

As illustrated in Figure 2, each experimental trial comprised three phases: FW, resting, and BW. In the FW phase, the participants stood on the treadmill, faced the treadmill’s control panel for 3 min, and then walked at a constant speed of 3 km/h for 4 min. Subsequently, they stood on the treadmill again for the final 3 min of the FW phase. Next, before proceeding to the BW phase, participants rested for approximately 10 min. Finally, by following the same procedure as the FW phase, participants were asked to turn around to face the rear end tennis ball, and thus reverse their walking direction for the BW phase of the experiment. Considering the potential danger of BW [69], by incrementally increasing the walking speed, the participants practiced BW prior to the actual experiments so that they could comfortably and gradually adapted to BW.

We compared the EEG signals measured before, during, and after walking. For both the FW and BW phases, signals from the middle 120 s (0.5 to 2.5 min) of the period before walking, the entire 240 sec of the walking period, and the initial 5–60 s of the period after walking were used in this study (Figure 3). The measurement period before walking (120 s) was considerably longer than that after walking (55 s) because our study focused on the effect of walking. Therefore, only the first minute of the period after walking was considered. Furthermore, the signals of the first 5 s of the period after walking were not used because the treadmill required a few seconds to completely stop.

### 2.3. EEG Signal Processing

A fundamental challenge of this study was the sensitivity of the EEG signals to environmental noise and motion artifacts. Therefore, we developed a systematic procedure to safeguard EEG signal fidelity. This subsection describes the adopted EEG signal processing procedure in a step-by-step manner.

#### 2.3.1. Preprocessing

During the preprocessing stage, the EEG signals were first subjected to bandpass filtering between 1 and 50 Hz.

#### 2.3.2. Least-Mean-Square Filter

Figure 4 depicts the basic structure of the least-mean-square (LMS) algorithm proposed by Widrow and Hoff [70], which has been extensively used in many signal processing applications because of its robustness and simplicity [71]. As illustrated in Figure 4, with the noise source *u*(*n*) as the input of an adaptive filter, the aforementioned algorithm attempts to develop a linear filter to simulate the dynamics of the noise path. In particular, by specifying error as the difference between the contaminated signal *d*(*n*) and the filter output *y*(*n*), the adaptive filter can be trained to minimize the sum of the mean square error. Because the noise source *u*(*n*) is independent of the signal source *s*(*n*), the filter output *y*(*n*) can only approximate the noise signal *v*(*n*). Consequently, the original signal can then be restored by subtracting the filter output *y*(*n*) from the contaminated signal *d*(*n*).

As illustrated in Figure 5, when implementing the LMS algorithm to eliminate gait-related motion artifacts, the GRF signal acquired from the force treadmill was selected as the noise source because the GRF is the only variable external force applied to the human body during walking. To determine the operating frequency of the adaptive filter, we first experimentally identified the bandwidth of the GRF signal. Analysis of the amplitude spectrum of the GRF signal acquired from the force treadmill revealed that the bandwidth of the GRF signal was marginally larger than 6 Hz. Therefore, the GRF and EEG signals were low-pass-filtered at 8.5 Hz and then down-sampled to 50 Hz. The down-sampling operation was introduced to reduce the computational cost.

As depicted in Figure 5, the output error *e*(*n*) of the adaptive filter was obtained as the difference between the low-passed, down-sampled, and contaminated EEG signal *d*(*n*) and the adaptive filter output *y_d_*(*n*). Next, by up-sampling the adaptive filter output to 250 Hz and subtracting the resulting signal *y_u_*(*n*) from the measured EEG signal *d*(*n*), we obtained the restored signal *r*(*n*). The restored signal *r*(*n*) was then visually inspected to remove apparently distorted signal segments. In specific, a signal segment was considered distorted and thus rejected if the amplitude or waveform of any signal channel looked different from those of the remaining signals. For example, the highlighted segment of Figure 6 was considered as an abnormal interval since the signal waveform of the P3 channel is different from the waveforms of the other channels. Apparently, visual inspection is a subjective process. Therefore, to safeguard the quality of the experimental data, our guideline was to sacrifice the false rejection error so that the false acceptance error can be minimized.

#### 2.3.3. ICA

ICA is a well-known, blind source, separation method and has been extensively used for EEG signal processing. By linearly decomposing the multichannel EEG data into a sum of maximally temporally independent statistical source signals through an unmixing matrix, ICA is used to identify independent components of artifacts and physiological signals. By using the ICA function of EEGLAB [72], we decomposed the nine-channel physiological signals (eight channels of EEG signals and one channel of ECG signals) into nine independent components to eliminate the influences of ECG, electrooculography, electromyography, and motion artifacts that had not been completely removed by the LMS filter. Specifically and similar to previous work (e.g., [73,74]), this study removed artifacts by visual inspection of the topography and time course of the ICA components. The EEG signals of the C3 and C4 channels were then reconstructed using retained ICA components.

#### 2.3.4. Statistical Outlier Removal

Figure 7 depicts the statistical outlier removal procedure employed in this study. The procedure involved dividing each channel of the EEG signals into multiple intervals, which are hereafter referred to as epochs. For the walking periods, an epoch was defined as the time interval required for two consecutive strides. These epochs were overlapped with a time interval of one stride. For the periods before and after walking, epochs were defined as 1-s sliding windows with a 0.5-s overlap.

In the time domain, the proposed statistical outlier removal procedure first involved determining the extreme value and kurtosis for each epoch. The extreme value of an epoch was defined as the maximum value of the absolute EEG signal value of that epoch. After calculating the mean and standard deviation of the extreme values of all epochs of an experimental trial, we could readily determine the Z-score for every epoch’s extreme value. As illustrated in Figure 7, an epoch was removed if the absolute value of the Z-score of its extreme value was larger than 3. In addition to the extreme value, an identical epoch removal rule was also implemented by comparing the kurtosis of the EEG signal amplitudes for every epoch.

In addition to the time domain features of extreme value and kurtosis, the aforementioned epoch removal rule was also applied to the 1–3 and 20–50 Hz spectral bands. In specific, an epoch was removed if the absolute value of the Z-score of the 1–3 Hz frequency band power spectral density curve’s extreme value was larger than 3. An identical epoch removal rule was also used for the 20–50 Hz frequency band. These two frequency bands were selected because the noise induced by electrooculography and electromyography mainly reside in these two frequency bands [75].

## 3. Results

Based on the data from C3 and C4 channels of the processed EEG signals, the results reported in this section are divided into three parts. The first part (Table 1) and second part (Table 2) indicate the effects of FW and BW on *μ*-rhythm suppression (desynchronization), respectively. The third part (Table 3) compares the effects of FW and BW on *μ*-rhythm suppression. For the statistical analyses performed in this study, all the data were log-transformed. All the hypotheses were one-directional, and their significance level was set as 0.05. Paired Student’s *t*-tests were used to detect statistically significant differences. Note that, based on the following two reasons, results associated with the other EEG channels are not reported in this work. Firstly, most comparative results of these channels were insignificant. Secondly, the focus of this work is on the mu rhythm which can only be assessed by C3 and C4 channels.

The symbols *BP*_0_, *DP*_0_, and *AP*_0_ are used to represent the mean *μ*_0_ rhythm’s relative energy for the periods before, during, and after the walking, respectively. Note that the relative energy is defined as the ratio of the energy of the frequency band of interest to the total energy of a signal. Similarly, *BP*_1_, *DP*_1_, and *AP*_1_ denote the mean relative energies of *μ*_1_ rhythm for the periods before, during, and after walking, respectively. Moreover, *BP*_2_, *DP*_2_, and *AP*_2_ denote the mean relative energies of *μ*_2_ rhythm for the periods before, during, and after walking, respectively. Note that the first letter of the above symbols is associated with walking periods (B for before, D for during, and A for after), and the subscript is used to represent rhythms (0 for *μ*_0_, 1 for *μ*_1_, and 2 for *μ*_2_).

To graphically demonstrate the overall *μ*-rhythm suppression effects of FW, Figure 8 depicts the average EEG signal amplitude spectra of the participants during the FW walking period (red lines) and before the FW walking period (green lines). Similarly, Figure 9 depicts the average EEG signal amplitude spectra of the participants after the FW walking period and before the FW walking period. Comparing these two figures, two interesting results were identified. First, as shown in Figure 8, walking apparently elevated the power of the EEG signals. This is also the reason why this study compared the relative energies of the *μ*_0_, *μ*_1_, and *μ*_2_ rhythms to try to identify the differences between BW and FW. Second, the overall *μ*-rhythm suppression effects were more pronounced during the walking period than after the walking period.

To quantitatively study whether the relative energies of *μ*_0_ rhythm, *μ*_1_ rhythm, and *μ*_2_ rhythm were significantly suppressed by FW during the walking period, the first set of alternative hypotheses tested was *DP*_0_ < *BP*_0_, *DP*_1_ < *BP*_1_, and *DP*_2_ < *BP*_2_. To study the desynchronization effects of the period after walking, the second set of alternative hypotheses tested was *AP*_0_ < *BP*_0_, *AP*_1_ < *BP*_1_, and *AP*_2_ < *BP*_2_. Table 1 summarizes the resulting *p*-values of the hypothesis testing results. When comparing their relative energies, the *μ*_1_ rhythm suppression effects were more pronounced in the period after walking than in the other periods, whereas the *μ*_2_ rhythm desynchronization effects were more prominent during the walking period than in the other periods (Table 1).

To graphically demonstrate the overall *μ*-rhythm suppression effects of BW, Figure 10 depicts the average EEG signal amplitude spectra of the participants during the BW walking period (red lines) and before the BW walking period (green lines). Similarly, Figure 11 depicts the average EEG signal amplitude spectra of the participants after the BW walking period and before the BW walking period. Similar to Figure 8 and Figure 9 of FW, it was found that BW elevated the EEG signal power and the overall mu-rhythm was more prominent during the walking period than after the walking period.

To quantitatively study the effects of BW on *μ*-rhythm desynchronization, the hypotheses tested in Table 1 for FW were repeated for BW. Only the walking period *μ*_1_ rhythm of the C4 channel was not significantly suppressed. By comparing the results in Table 1 and Table 2, we found that FW and BW shared two similar suppression patterns: (1) the *μ*_1_ rhythm suppression effects were more prominent in the period after walking than during walking and (2) the *μ*_2_ rhythm suppression effects were more pronounced during walking than in the period thereafter. A distinct difference between FW and BW was that in the period after walking, the *μ*_2_ rhythm suppression effects were significant for BW and nonsignificant for FW.

With Table 1 and Table 2 demonstrating the impacts of FW and BW, respectively, the next step is to directly compare the impacts of FW and BW. To more rigorously compare the suppression effects of FW and BW on *μ*-rhythm, *μ*_1_ rhythm, and *μ*_2_ rhythm with the baseline of the period before walking, we performed two sets of hypothesis tests. By assuming that BW had larger *μ*-rhythm suppression effects than FW, the first set of alternative hypotheses was that the *BP*_0_ − *DP*_0_, *BP*_1_ – *DP*_1_, and *BP*_2_ – *DP*_2_ values for BW were larger than those for FW. Similarly, for the period after walking, the second set of hypotheses was that the *BP*_0_ − *AP*_0_, *BP*_1_ – *AP*_1_, and *BP*_2_ – *AP*_2_ values for BW were larger than those of FW.

Table 3 summarizes the *p*-values obtained in the aforementioned hypothesis tests. All the differences between the *μ*_1_ rhythm suppression effects of FW and BW were nonsignificant. However, Table 2. rhythm hypothesis test results in Table 3 indicated that the *μ*_2_ rhythm suppression effects of BW were significantly larger than those of FW. The results of Table 3 show that *μ*_1_ rhythm is relatively insensitive to walking direction. By contrast, the *μ*_2_ rhythm suppression effect changes significantly with the walking direction. As a result, the relative energy of the *μ*_2_ rhythm can represent a distinct measure that can be used to differentiate the impacts of FW and BW. These results suggest that *μ*_1_ rhythm and *μ*_2_ rhythm are functionally different.

## 4. Discussions

Before further exploring the implications of the summarized results, we first examined whether our results were consistent with the three major *μ*-rhythm properties mentioned in Section 1.3. The first property is that cognition and motor tasks can lead to *μ*-rhythm desynchronization. Our results were clearly consistent with this property because most of the results for *μ*-rhythm suppression tests, summarized in Table 1 and Table 2, were statistically significant.

The second *μ*-rhythm property is that *μ*_1_ rhythm and *μ*_2_ rhythm are functionally different. This property is also supported by the results presented in Table 1, Table 2 and Table 3 because the three tables list fundamentally different statistical test results for *μ*_1_ rhythm and *μ*_2_ rhythm. For example, after comparing the suppression effects of FW and BW, the four *μ*_1_ rhythm statistical tests failed to provide significant results, whereas the four *μ*_2_ rhythm statistical tests yielded significant results (Table 3, final two columns). Similar disparate significance results for *μ*_1_ rhythm and *μ*_2_ rhythm have also been observed in related studies. For instance, [47] found that only *μ*_2_ rhythm was significantly suppressed during preparation for finger movement. Moreover, [76] found that 30 min of passive high-frequency repetitive sensory stimulation led to significant *μ*_2_ rhythm suppression. The data obtained from observation and anticipation experiments regarding the tennis-related actions of experienced and inexperienced players indicated that the experienced players exhibited earlier and greater *μ*_2_ rhythm desynchronization than the inexperienced players did [61]. Notably, in all the aforementioned studies, *μ*_1_ rhythm has not been found to be significantly desynchronized. Consequently, our experimental results clearly support the notion that *μ*-rhythm desynchronization is not a unitary phenomenon and that *μ*_1_ rhythm and *μ*_2_ rhythm are functionally different. As a result, compared to studying only the *μ*_0_ rhythm, more information can be gained by studying both *μ*_1_ and *μ*_2_ rhythms. At the same time, the *μ*_0_ rhythm can be used to characterize the overall response of the sensorimotor alpha rhythm.

The third most observed *μ*-rhythm property is that the degree of *μ*-rhythm suppression is task-dependent. For example, [47] revealed that finger movement resulted in larger *μ*-rhythm suppression effects than foot movements, and [48] suggested that increasing task complexity resulted in a high level of suppression. Relative to passive motor tasks, active motor tasks cause larger *μ*-rhythm desynchronization effects [36,65]. When providing an engaging feedback environment, the three-dimensional visualization of movements resulted in larger *μ*_2_ rhythm suppression effects than two-dimensional visualization did [77]. Observation and anticipation experiments regarding tennis-related actions indicated that experienced players had earlier and greater *μ*_2_ rhythm desynchronization effects than inexperienced players did [61]. This result was explained by the observation that the motor expertise of the experienced players helped them understand their opponent’s intention; therefore, they were cognitively more engaged than the inexperienced players. Similarly, a simulated aircraft landing study revealed that *μ*-rhythm was significantly more suppressed in pilots than in novices because the strength of *μ*-rhythm suppression increased with the sense of presence experienced by the experienced pilots [78]. These results suggest that higher levels of human engagement can result in a larger *μ*-rhythm response. These results have critical implications for rehabilitations because favorable rehabilitation outcomes appear to be strongly associated with high patient engagement [79,80].

By identifying the differences between FW and BW, our results successfully indicated that *μ*-rhythm suppression effects are task-dependent. Specifically, our experimental results demonstrate that BW induced larger *μ*-rhythm suppression effects than FW. Consequently, our results appear to provide support for the notion that BW can provide potential benefits over FW in gait rehabilitation. However, a fundamental limitation of the current study is that the results still did not provide direct evidence to establish a causal link between *μ*-rhythm suppression and a rehabilitation outcome. However, several results have emerged to support the existence of such a cause-effect relationship. For example, a 30-min *μ*-rhythm neurofeedback training session could directly affect motor-cortical plasticity [81] and facilitate the early acquisition of a procedural motor task [82]. When reinforcing successful *μ*-rhythm desynchronization with robotically assisted hand manipulation, patients with chronic stroke exhibited an improvement of 3.4 points on the upper-limb Fugl–Meyer scale [83]. Another study revealed that *μ*_2_ rhythm-based neurofeedback training had positive effects on memory functions and led to neuronal plasticity processes in chronic stroke victims [84].

To further clarify the associations between *μ*-rhythm, neuroplasticity, and rehabilitation, future studies should perform a sufficient number of longitudinal rehabilitation experiments to completely verify this cause–effect relationship between the intensity of *μ*-rhythm suppression and a rehabilitation outcome. Specifically, to more comprehensively compare the effects of FW and BW on the brain, future studies should increase the number of EEG channels to cover additional brain regions. Moreover, the potential suppression effects associated with other frequency bands, such as beta and gamma rhythms, can also be investigated. Considering the benefits of neurofeedback, future studies could develop a *μ*-rhythm-based BW training neurofeedback system for gait rehabilitation. If patients can observe the pattern and intensity of their own *μ*-rhythm response in real time, they might become more engaged in the rehabilitation process, which could result in improved rehabilitation outcomes.

In summary, this work demonstrates that walking leads to *μ*-rhythm suppression but the degree of *μ*-rhythm suppression is task-dependent. In addition, by showing that *μ*_1_ rhythm and *μ*_2_ rhythm are functionally different, a primary finding of this study is that the *μ*_2_ rhythm suppression effects of BW are significantly larger than those of FW. Since a larger *μ*_2_ rhythm response is associated with higher levels of human engagement and favorable rehabilitation outcomes appear to be strongly associated with high patient engagement, our results appear to provide support for the notion that BW can provide potential benefits over FW in gait rehabilitation. The fundamental limitation of this work is that the results of this study are still not able to prove the causal link between *μ*-rhythm suppression effects and rehabilitation outcome. Finally, it should be noted that the impacts of locomotion on EEG signals are not limited to the 8–12 Hz frequency bands [42,55,56,57]. Extending this work by studying the associations between treadmill locomotion and other frequency bands of the EEG signals is a possible and valuable future work.

## 5. Conclusions

Although the differences and similarities between FW and BW have been comprehensively studied, current evidence for the rehabilitation benefits of BW over FW remains inconclusive. To explore the potential of BW for the treatment of people with gait impairments, we differentiated the brain responses induced by FW and BW by comparing their *μ*-rhythm suppression effects. Toward this goal, the 8–12-Hz frequency band of *μ*-rhythm was divided into the 8–10-Hz sub-band of *μ*_1_ rhythm and the 10–12-Hz sub-band of *μ*_2_ rhythm.

In agreement with previous results, our experimental results indicated that *μ*-rhythm can be suppressed by FW and BW. A primary finding of this study is that the *μ*_2_ rhythm suppression effects of BW are significantly larger than those of FW. By contrast, the *μ*_1_ rhythm suppression effects of FW and BW were not significantly different. According to the aforementioned results, future research could involve a longitudinal walking rehabilitation study to investigate the causal link between *μ*_2_ rhythm response and rehabilitation outcomes. Another area for future research is the development a neurofeedback system that monitors the pattern and intensity of *μ*_2_ rhythm responses to improve the effectiveness of walking rehabilitation.

## Figures and Tables

**Figure 1 sensors-20-07250-f001:**
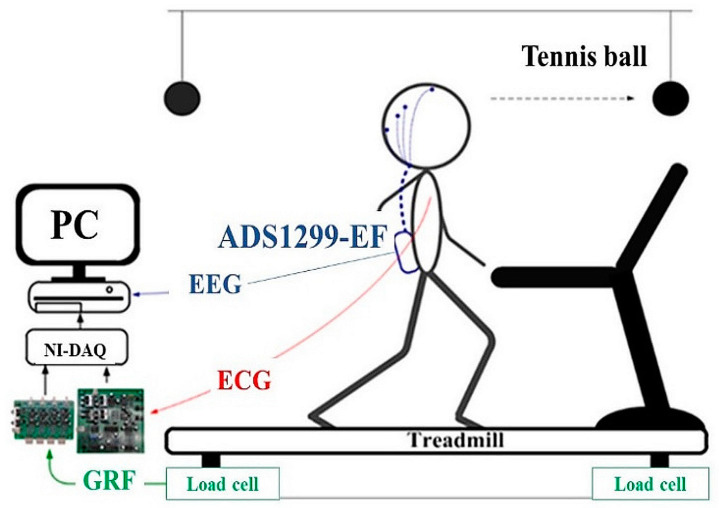
Illustration of the experimental system.

**Figure 2 sensors-20-07250-f002:**
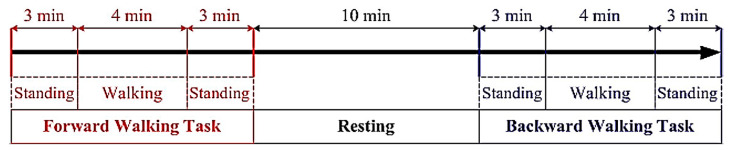
Experiment procedure.

**Figure 3 sensors-20-07250-f003:**
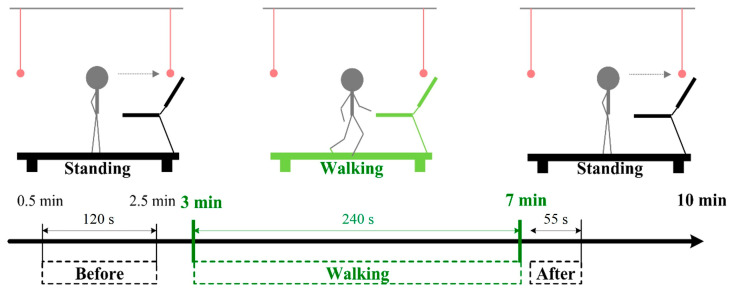
Signal periods used in this study.

**Figure 4 sensors-20-07250-f004:**
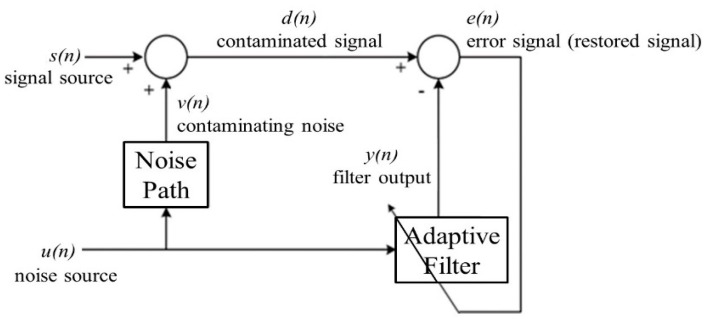
Basic structure of the LMS-based adaptive filter.

**Figure 5 sensors-20-07250-f005:**
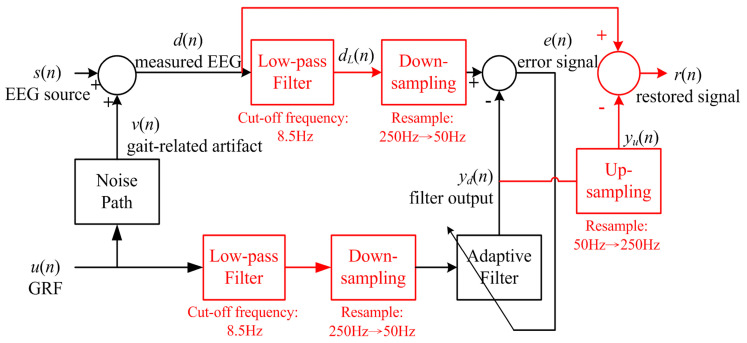
Block diagram of the LMS filter developed in this study.

**Figure 6 sensors-20-07250-f006:**
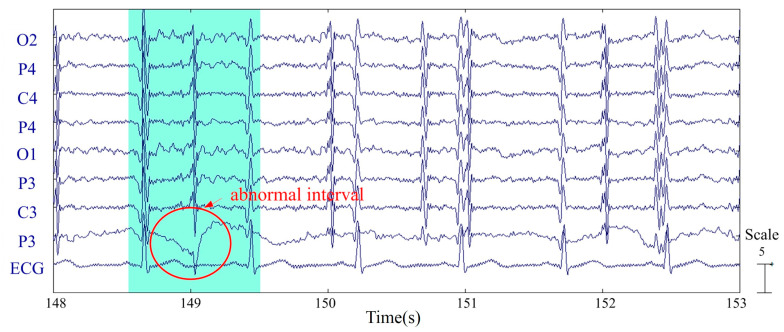
Remove a distorted signal segment by visual inspection.

**Figure 7 sensors-20-07250-f007:**
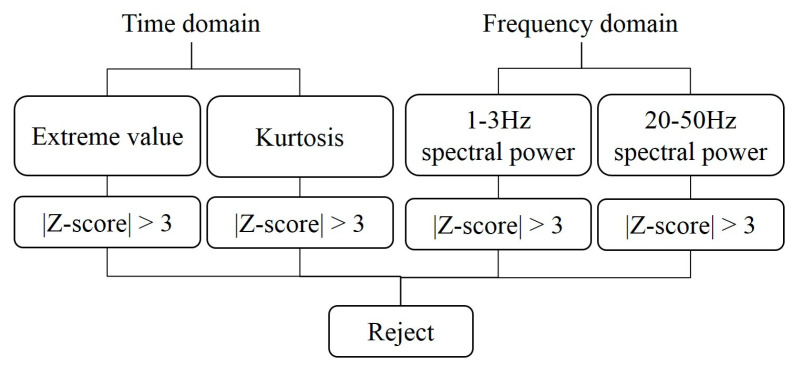
Proposed statistical artifact removal procedure.

**Figure 8 sensors-20-07250-f008:**
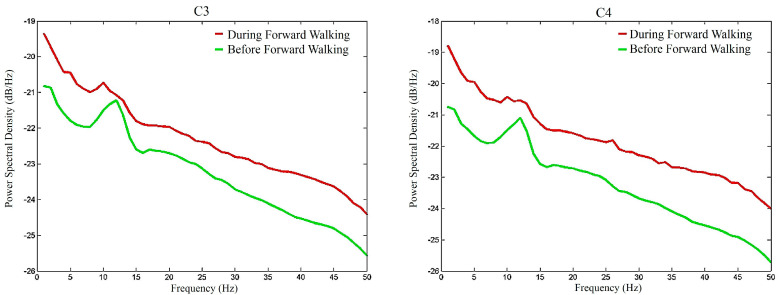
The amplitude spectra of the C3 and C4 EEG electrode signals during and before the forward walking period.

**Figure 9 sensors-20-07250-f009:**
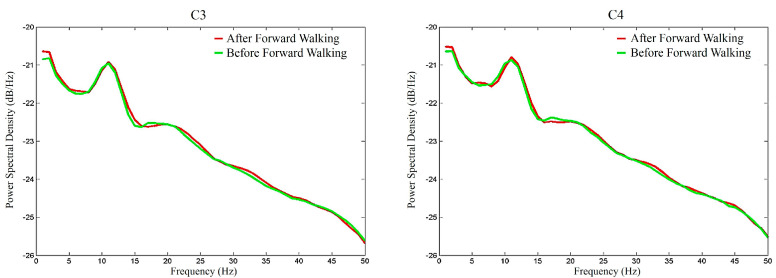
The amplitude spectra of the C3 and C4 EEG electrode signals after and before the forward walking period.

**Figure 10 sensors-20-07250-f010:**
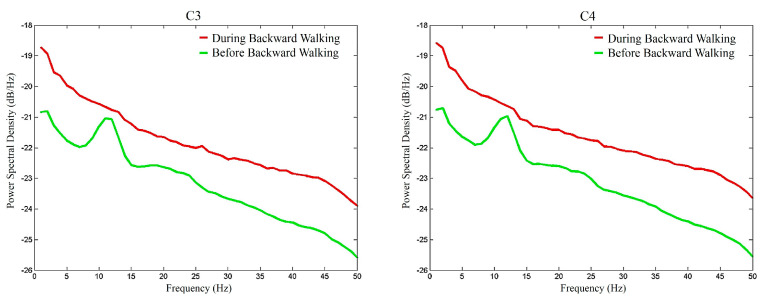
The amplitude spectra of the C3 and C4 EEG electrode signals during and before the backward walking period.

**Figure 11 sensors-20-07250-f011:**
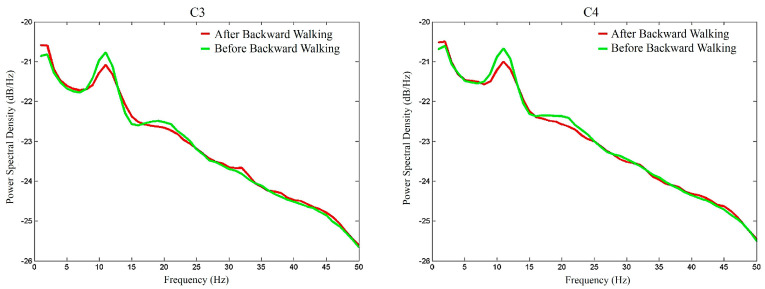
The amplitude spectra of the C3 and C4 EEG electrode signals after and before the backward walking period.

**Table 1 sensors-20-07250-t001:** Summary of the *p*-values when comparing the *μ*-rhythms in different FW periods.

Walking Periods	Channels	Frequency Bands		
		*μ* _0_	*μ* _1_	*μ* _2_
During vs. Before	C3	0.006	0.136	0.003
	C4	*	0.016	*
After vs. Before	C3	0.040	0.020	0.123
	C4	0.076	0.006	0.310

* *p*-value < 0.001.

**Table 2 sensors-20-07250-t002:** Summary of the *p*-values when comparing the *μ*-rhythms in different BW periods.

Walking Periods	Channels	Frequency Bands		
		*μ* _0_	*μ* _1_	*μ* _2_
During vs. Before	C3	*	0.032	*
	C4	*	0.063	*
After vs. Before	C3	0.003	0.005	0.006
	C4	*	0.001	0.003

* *p*-value < 0.001.

**Table 3 sensors-20-07250-t003:** Summary of the *p*-values when comparing the *μ*-rhythms suppression effects of FW and BW.

Walking Periods	Channels	Frequency Bands		
		*μ* _0_	*μ* _1_	*μ* _2_
During	C3	0.052	0.117	0.022
	C4	0.089	0.370	0.012
After	C3	0.016	0.218	0.013
	C4	0.009	0.471	0.004

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
