# Peer review of "Backward Walking Induces Significantly Larger Upper-Mu-Rhythm Suppression Effects Than Forward Walking Does"

_sensors, 2020, doi:10.3390/s20247250_

Round 1

Reviewer 1 Report

  1. In Figure 1, two tennis balls are shown, one in front of a forward walking subject and one behind. Please clarify because it isn't properly discussed in the text, whether the subject was required to turn round and face the 'behind' tennis ball to do the backwards walking and whether doing backwards walking on a treadmill is potentially dangerous. (The reviewer has never been on a treadmill and so wouldn't know and the same may possibly be the case for readers of the paper).
  2. It is not explained in the description of the experiment why nine electrodes are used and then only the signals from C3 and C4 are ultimately processed and used to calculate the final result. This is more especially the case when it is noted in lines 104-5 that the study developed adaptive filters for every EEG channel. If the other channels gave no useful signals in the mu bands or gave negative results on the hypotheses tested, then this should be reported.
  3. In Section 2.3.3, line 251 should that be " and motion artifacts that had not been completely removed " since you do not need to remove them if they have been completely removed already?
  4. In line 252, do they mean "Specifically and similar to previous work (e.g...)"
  5. In line 254,  it should read: " were then reconstructed using ..." ​
  6. Regarding the mentions in line 241-2 and lines 252-3 where the reference is made to 'removal by visual inspection' or adjustments made after visual inspection, it might also be helpful if some extra information e.g. a figure was given to show what these are.
  7. In Section 2.3.4 about statistical outlier removal, what criterion exactly apart from abs(z-score) > 3 is applied to the frequency bands epoch removal rule. Does that mean Z-score of the power level in those indicated bands?
  8. On lines 266-7 it is stated that 'an epoch was removed if the Z-score of its extreme value was larger or smaller than 3', but this isn't really the case as that would remove all epochs! It is also not what is shown in Figure 6. What is presumably meant is if the absolute value of the Z-score is larger than 3 or if the Z-score is larger than 3 or less than -3 and so this sentence should be corrected.
  9. DP_0, BP_2 etc. are meant to signify the particular mu rhythm's 'relative energy' in particular periods, but relative to what? We really need to see a more rigorous definition of this quantity and how it is calculated.
  10. It is not clear that the description of the 'additional hypothesis tests' and Table 3 really adds anything to the analysis. It is self-evident from Tables 1 and 2 that the difference of these two would result in something looking like Table 3 and if these are post-hoc hypotheses as seems to be suggested from the way lines 309-315 reads then maybe that table should not be included.
  11. It would also be useful to have a clear discussion of the role of Mu-0. It shows up in the tables and in the results but does not seem to be as discussed as mu-1 and mu-2.
  12. While it is clear that statistical tests have been used to establish the relationship between Mu rhythm suppression and forwards/backwards walking,  it is surprising not to see any actual examples of the signals actually recorded and their processing stages included in the results.
  13. A large part of the discussion still seems to be discussing other people's results rather than discussing the results or other findings from this experiment which could also include discussion of any null or negative findings.
  14. line 367:  should read 'several'

Reviewer 2 Report

This study compared human EEG signals while subjects walked either forward or backward, and also described a novel method to remove motion artifacts from the collected signals. The data have relevance to normal motor function, perhaps as well to gait rehabilitation practice, and the methods employed are sound. This will be a useful contribution. But the authors need to explain better the motivation for the study, clarify some methodological details, actually show some of their data, and temper their conclusions.

(1) The central hypothesis of the study, and relevance to rehabilitation are unclear.

The stated aim of the study is to characterize different EEG signals during forward and backward walking. “The hypothesis of this study is that the brain responses associated with BW is different from those associated FW.” (should be "…from those associated with FW…”) This is not a very compelling hypothesis. Should we be surprised that different motor behaviors are correlated with different EEG signals? And how does this relate to gait rehabilitation? To restate those questions: Why is a change in EEG activity not merely a correlate of different motor behaviors, and what about the EEG differences between BW and FW conditions will guide rehabilitation?

    Line 69 “…to use EEG to compare the effects of FW and BW…”  This is poorly phrased, and again reveals a fundamental ambiguity in the motivation for the study. Are changes in EEG signals an effect (i.e. a result) of forward or backward walking, or do the different EEG signals seen in the two conditions arise because the different brain signals CAUSE different walking patterns. Fundamentally, the authors are describing differences in EEG signals in 2 conditions. This is worthwhile to document, but the logic (and thus interpretation) is unclear.

    Last paragraph of intro line 148-160 is confusing. There is not a clear statement of the logic of the study. Why should mu-suppression have anything to do with rehabilitation effectiveness, and how will data from this study lead to changes in planning BW training?

Line 129-130 how are they different? Even one sentence clarification here would be useful

Lines 136-142, likewise not clear what “movement-type specific pattern” means. Important to clarify since here is motivation for studying different frequency components.

Lines 157-170 of Introduction should be in Discussion not at end of Intro.

2) Some methodological details are lacking.

Line 119 refers to “sensorimotor cortex”, but authors should specify which cortical (Brodmann) areas are likely to be contributing to different electrodes’ signals. Are they revealing signals related to motor processing, sensory processing, or something else?

A 0.05 significance level is not strict enough, and they should state if they corrected for multiple comparisons (which they should of course do).

Method line 241 “The restored 241 signal r(n) was then visually inspected to remove apparently distorted signal segments.”  What constitutes distorted, and how did the authors choose such segments. It seems quite subjective and arbitrary to remove portions of data based on visual inspection, when a key aspect of the study is the noise-removal algorithm they employed.

The authors should show some raw EEG traces together with cleaned-up versions to document what they mean by “distorted segments.”  What does distorted mean? This relates to my comment below that the authors show none of their EEG signal data, which they must do.

Line 262 what is the measure whose kurtosis or distribution were analyzed, just distribution of EEG peak voltage values? This needs to be clarified

3) The authors MUST show some of their actual data. None are shown.

The authors should show some EEG traces to illustrate mu-suppression during FW vs BW, and the validity of their artifact removal algorithm.

    It is also somewhat unclear what were the actual response measures being plotted. It would help to plot graphs of their summary data rather than merely showing results as tables so readers can SEE the difference. Furthermore, tables report only the p-values not the magnitude of the effect, nor was it clear what parameter they were describing (was it the relative power of the mu band for example?)

    A core point of the paper is to compare FW vs BW, but these are plotted in different tables. The reader needs to look at different tables to compare these. Instead, an individual figure should compare a particular metric in the FW and BW conditions.

    The text never states why some numbers in the tables are red vs black (significant vs insignificant but not explained).

    Why do the authors show data in the Tables only for C3 and C4 channels? The authors need to state why these channels’ data were selected.

(4) Discussion left me confused. Did not effectively summarize data to convince me they had addressed all the issues they raised in the Introduction.

For example, they report that their results show mu1 and mu2 components are different since they behave differently under different motor conditions. But what this really means, and how this relates to gait rehabilitation are still unclear. For example on line 363 they state “Consequently, our results appear to provide support for the notion that BW can provide potential benefits over FW in gait rehabilitation.” This does not follow at all.

(5) Minor errors

FW and BW should be defined on 1st use (for ex line 43)

Line 31 “that” is repeated

Line 51 “specific” or line 55 “In specific” should be “Specifically” (or could simply be deleted)

Line 58 “has more intensive cortical participation” what does this mean?

Line 61,67 “potentials” should be “potential”

Line 124 “decreases” should be “decrease”

Line 150 “is” should be “are”

Line 238 “he” should be “the”

Line 254 “reconstrued” should be “reconstructed”

Round 2

Reviewer 2 Report

The authors addressed all of the issues I raised in my initial review. The manuscript is much improved with addition of requested clarifications and better documentation of data. This is a useful contribution.